# Molecular Mechanisms Linking Osteoarthritis and Alzheimer’s Disease: Shared Pathways, Mechanisms and Breakthrough Prospects

**DOI:** 10.3390/ijms25053044

**Published:** 2024-03-06

**Authors:** Idiongo Okon Umoh, Helton Jose dos Reis, Antonio Carlos Pinheiro de Oliveira

**Affiliations:** Departamento de Farmacologia, Instituto de Ciências Biológicas, Federal University of Minas Gerais, Av. Antonio Carlos 6627, Belo Horizonte 31270-901, MG, Brazil; uidiongo@yahoo.com

**Keywords:** neurodegeneration, peripheral inflammation, neuroinflammation, cytokines, cognition, arthritis

## Abstract

Alzheimer’s disease (AD) is a progressive neurodegenerative disease mostly affecting the elderly population. It is characterized by cognitive decline that occurs due to impaired neurotransmission and neuronal death. Even though deposition of amyloid beta (Aβ) peptides and aggregation of hyperphosphorylated TAU have been established as major pathological hallmarks of the disease, other factors such as the interaction of genetic and environmental factors are believed to contribute to the development and progression of AD. In general, patients initially present mild forgetfulness and difficulty in forming new memories. As it progresses, there are significant impairments in problem solving, social interaction, speech and overall cognitive function of the affected individual. Osteoarthritis (OA) is the most recurrent form of arthritis and widely acknowledged as a whole-joint disease, distinguished by progressive degeneration and erosion of joint cartilage accompanying synovitis and subchondral bone changes that can prompt peripheral inflammatory responses. Also predominantly affecting the elderly, OA frequently embroils weight-bearing joints such as the knees, spine and hips leading to pains, stiffness and diminished joint mobility, which in turn significantly impacts the patient’s standard of life. Both infirmities can co-occur in older adults as a result of independent factors, as multiple health conditions are common in old age. Additionally, risk factors such as genetics, lifestyle changes, age and chronic inflammation may contribute to both conditions in some individuals. Besides localized peripheral low-grade inflammation, it is notable that low-grade systemic inflammation prompted by OA can play a role in AD pathogenesis. Studies have explored relationships between systemic inflammatory-associated diseases like obesity, hypertension, dyslipidemia, diabetes mellitus and AD. Given that AD is the most common form of dementia and shares similar risk factors with OA—both being age-related and low-grade inflammatory-associated diseases, OA may indeed serve as a risk factor for AD. This work aims to review literature on molecular mechanisms linking OA and AD pathologies, and explore potential connections between these conditions alongside future prospects and innovative treatments.

## 1. Alzheimer’s Disease: Pathogenesis and Epidemiology

### 1.1. Pathogenesis

Alzheimer’s disease (AD) is a multifaceted, complex type of neurodegenerative disorder defined by the gradual deterioration in a patient’s cognitive abilities alongside numerous intricate molecular events in neuronal brain cells such as neuroinflammation, accumulation and aggregation of amyloid-β (Aβ) and hyperphosphorylated TAU, respectively, in selective and distinctive varied brain regions and neuron types ultimately leading to synaptic damage, loss and dysfunction. This is followed by neuronal death and ultimately progressive cognitive decline [1,2,3]. Below, the mechanisms involved that cause synaptic dysfunction and subsequent loss of neurons in AD are explained.

#### 1.1.1. Neuroinflammation

AD is also characterized by long-term inflammation in the brain involving activation of microglial cells, invasion and activated peripheral immune cells (monocytes and T cells), along with generation and release of proinflammatory mediators that lead to disease progression [4,5,6].

#### 1.1.2. Amyloid-β Hypothesis

Accumulation of amyloid beta (Aβ) containing plaques in different brain areas is central to the amyloid hypothesis. Neuronal extracellular deposits of this peptide are generated from the amyloidogenic pathway by the cleavage of amyloid precursor protein (APP) by beta- and gamma-secretase enzymes. The various insoluble species, especially oligomers, disrupt normal synaptic functions and trigger neurotoxic events in the neurons, leading to a decline in cognitive functions of the brain [7]. Aβ peptide accumulation, in particular, is due to dysregulation or imbalance in the synthesis and removal of accumulated Aβ oligomers, which resulted from the hydrolytic actions of beta-site APP cleaving enzyme-1 (BACE-1) with γ-secretase cleavage enzymes on APP; some mechanisms involved in Aβ clearance include ubiquitin–proteasome system (UPS), autophagy, apolipoprotein-E processes, proteolytic regulation and clearance of blood–brain barrier (BBB) [8].

#### 1.1.3. TAU Pathology

The complex involvement of TAU pathology is of deep significance in neurophysiology. Intracellular aggregation of abnormally phosphorylated TAU leads to the formation of neurofibrillary tangles (NFTs), a hallmark of AD pathology. Recent studies have highlighted the multifaceted part played by hyperphosphorylated TAU in neuroinflammation, synaptic dysfunction and impairment and the overall neurodegeneration leading to the progressive cognitive decline observed in AD patients [9,10]. Toxic Aβ produced by mutated APPs also triggers TAU pathology through a cascade of events involving synaptic impairment, neuroinflammation and neuronal damage, which ultimately leads to the formation of NFTs [9,11,12].

#### 1.1.4. Dysregulated miRNAs

Several non-coding RNAs have appeared in proposed biomarker lists for neurodegenerative conditions. Among the most promising candidates is microRNA (miRNAs), a potent regulator of gene expression [13,14,15,16]. miRNAs are relatively short molecules (20 to 22 nucleotides in length) that can degrade or suppress the translation of their many complementary mRNA targets within a cell but only through the tissue type involved [16]. Endogenous miRNAs are often found at their target sites in the 3′UTR region of mRNA, where they form an imperfect duplex hybrid and interfere with translation [17]. The function of miRNA is not limited to the cell from which it was produced. Many extracellular miRNAs are released, and exchanged in communication between blood, cerebrospinal fluids (CSF), brain and periphery [18]. Research has shown several miRNAs dysregulated in human and animal models of neurodegenerative diseases, supporting their role as disease biomarkers [14]. These miRNAs regulate key genes and signaling pathways involved in the amyloidogenic pathway, Aβ clearance, tau hyperphosphorylation and aggregation, which could influence AD-related pathways [19]. Analysis of human AD *post-mortem* tissues and circulating fluids found dysregulation in miRNAs levels, such as miR-7, 9, 16, 29a, 29b, 32, 34a, 34c, 101, 124, 125b, 128, 132, 135a, 146a, 195 and miR-218 [19]. miR-9, 29a, 29b, 124, 135a and miR-195 may also influence the amyloidogenic pathway through downregulation of beta-secretase enzyme 1 (*BACE1)*. Dysregulation of miR-7, 9, 16, 34a and miR-101 may impair Aβ clearance. TAU protein levels and/or phosphorylation are also affected by dysregulated miR-16, 124, 125b, 132 and miR-218. Finally, miR-16, 124, 125b and miR-132 affect both Aβ or tau pathways [19,20,21,22].

#### 1.1.5. Vascular Factors

Current research suggests that conditions such as hypertension, obesity and clot formation in the cerebral capillaries which predisposes to vascular damage may contribute to AD pathogenesis by impairing brain blood flow to certain brain parts, resulting in neuronal death and cognitive decline in AD patients [23,24,25,26].

#### 1.1.6. APOE and Amyloid Metabolism in the Brain

Produced by the astrocytes in the CNS, APOE (E2, E3) assists in the removal of Aβ oligomers, thereby preventing the formation and a relative overabundance and accumulation of the Aβ_1-42_ species. Nevertheless, APOE-ε4 cannot render the peptide harmless by removing it from or breaking it down within the central nervous system as the other forms of APOE. This allows for the accumulation of Aβ species that ultimately results in cell death and dementia. Therefore, expression of APOE-ε4 serves as a genetic risk factor for late-onset AD [27,28].

#### 1.1.7. Trisomy 21—Down’s Syndrome

The APP gene is found on chromosome 21. Persons with Down’s syndrome suffer early onset of AD because they have an extra copy of chromosome 21, which expresses APP in greater amounts.

#### 1.1.8. Genetics in EOFAD and LOAD

Genetic mutations in APP and PSEN1/PSEN2 genes play crucial roles in familial cases, i.e., early-onset familial AD (EOFAD), a condition that occurs before 65 years of age. On the other hand, late-onset AD (LOAD), more prevalent in patients older than 65 years of age, occurs mainly as a result of interaction between genetic risk factors (APOE-ε4, APP, PSEN1, PSEN2, ABCA7 (ATP-binding cassette transporter A7), CLU (Clusterin), CR1 (Complement Receptor 1), TREM2 (Triggering Receptor Expressed on Myeloid Cells 2), SORL1 (Sortilin-related receptor), TOMM40 (Translocase of Outer Mitochondrial Membrane 40) [28,29,30,31,32,33] and non-genetic risk factors like diet, smoking, lack of education, head injury, metabolic and inflammatory diseases which also may contribute to the development of AD [34,35,36,37,38]. These risk factors show us precisely how genetics, lifestyle and environmental factors are so intimately connected to the etiology of LOAD. This interaction of genetic and environmental factors leads to a chain of events beginning with the accumulation of Aβ plaques and tau protein abnormalities, followed by inflammation and neuronal damage, eventually leading to the typical cognitive decline of AD.

### 1.2. Epidemiology of AD

From an epidemiological perspective, AD primarily affects the elderly with age itself being a primary risk factor [39]. It is believed that some 50 million people around the world have this condition and that as the global population ages, these numbers will continue to rise sharply [40].

#### 1.2.1. Global Prevalence

AD is a global health problem, representing 60–70% of the cases of dementia that affects people of all races or ethnicities, with about 60% of these cases living in low- and middle-income countries. Unfortunately, there are about 10 million new cases of dementia every year. With an increase in the aging population worldwide, it is estimated that, by 2050, the total number of people with AD will reach 152 million [41,42,43,44].

#### 1.2.2. Global Variation

Due to differences in genetics, lifestyle, access to medical care and socio-economic factors alongside the complex links between lifestyle factors and genetic predispositions, the incidence rate for AD also differs from place to place. Emerging evidence in research has found lower prevalence in Mediterranean countries that may be linked to or due to the protective effects of the diet [45,46]. High-income countries often bear a substantial burden of the disease due to an increasing aging population [35,41]. Delving into these subtle differences in the global epidemiological landscape provides valuable insights into future research and public health strategies for combating AD.

#### 1.2.3. Gender Disparities

Regarding the prevalence, AD is more prevalent in women than men with the involvement of composite and multifaceted underlying factors. This difference is related to a greater extent to biological differences such as hormonal changes, genetic predispositions and longevity, which contribute to the susceptibility to the disease [47].

## 2. Peripheral Inflammation and Neurodegenerative Diseases: Insights into AD Pathogenesis

Immunological processes within the brain are key factors for the pathogenesis of AD. Aggregated proteins bind to pattern recognition receptors in microglial and astroglial cells and trigger an innate immune response involving the release of inflammatory molecules that contribute to the progression and severity of disease [4,48]. Interestingly, by using genome-wide analyses, several genes that increase the risk of developing sporadic AD have already been identified [4]. Some of these genes are required for controlling glial cell clearance of misfolded proteins; others play a role in modulating inflammation [4]. Also, many external factors including systemic inflammation and obesity can affect the immune processes of the brain, making disease progress even worse. Approaches that involve modifying these genetic risk factors and targeting involved immune mechanisms point to possible future AD prevention strategies and potential therapies [4,5,49]. The following section presents recent research highlighting the complicated relationship between peripheral inflammation and AD pathogenesis.

### 2.1. The Role of Inflammation in AD Pathogenesis

#### Systemic Inflammation and Blood–Brain Barrier Dysfunction

Emerging studies have uncovered how systemic inflammation can threaten changes in the blood–brain barrier (BBB) and contribute to the progression of neurological conditions such as AD and multiple sclerosis [50,51]. Under these conditions, systemic inflammation compromises the BBB by upregulating levels of pro-inflammatory mediators (TNF, IL-1β). This breakdown of BBB allows peripheral inflammatory substances or cells to diffuse or migrate into the brain parenchyma [52,53]. The resulting traffic of immune cells and peripheral inflammatory mediators affects the BBB and its neurovascular components, prompting physiological disturbances that contribute to neuronal injury and cognitive declines in AD (51). Additionally, peripheral inflammatory mediators, which participate in chronic inflammatory conditions, may trigger glial activation that amplifies neuroinflammation which worsens BBB disruption and exacerbation of neurodegenerative processes [54,55].

Nation et al. [56], by using novel early biomarker detection techniques, showed that individuals with early cognitive decline/dysfunction develop BBB breakdown in the hippocampus notwithstanding Aβ or TAU changes. In a comprehensive review by Zhao et al. [57], the authors discussed increasing research evidence emphasizing the implications of BBB dysfunction in neurological disorders and complex multifactorial diseases, including AD. However, the authors finally highlighted that the relationship between factors such as neurovascular integrity, functional and structural brain connectivity and neurological symptoms awaits to be directly and precisely dissected in in vivo studies [57].

### 2.2. Peripheral Inflammation and AD Risk Factors

#### 2.2.1. Genetic Susceptibility

Recently, research has identified genetic factors associated with both risk of AD and immune function [58]. AD risk and immune function have both been affected and influenced by specific genetic variants such as APOE, TREM2 (Triggering Receptor Expressed on Myeloid Cells 2), CD33 (Cluster of Differentiation 33) and MS4A (Membrane-Spanning 4-Domains Subfamily A), the detection of which has emphasized their importance in determining microglial behavior and influencing Aβ processing [59,60,61,62]. Such findings reveal the complex interplay between genetics and peripheral inflammation in AD etiology.

#### 2.2.2. Environmental Factors

Environmental factors, including diet, physical activity and pollution exposure can affect peripheral inflammation, which could, in turn, promote LOAD [63,64]. For example, diet is important to the risk of AD; it has been observed that a Western-style high-saturated fat and cholesterol diet promotes deposition of Aβ plaque formation, while antioxidant-rich diets such as those containing vegetables and fruits may prevent the aggregation of Aβ peptides in the brain [65,66]. Lifestyle modifications might be one way to make contributions toward AD prevention (avoiding or delaying).

Research on animal models and humans has illustrated that regular exercise decreases the chances of developing AD, likely through improving cerebral blood flow and increasing neurogenesis and Aβ clearance [67,68]. The neuroprotective effects of exercise point to its potential as a non-pharmacological approach to preventing AD.

Finally, some environmental pollutants containing fine particulate matter and heavy metals can accumulate in tissues, and represent a risk for developing AD through mechanisms that involve oxidative stress, neuroinflammation and TAU hyperphosphorylation [69,70,71]. These findings emphasize the greater need for stricter environmental regulations as one of the ways to reduce the risks of AD.

#### 2.2.3. Activation of Microglia

Prolonged peripheral inflammation may lead to microglial activation, causing chronic neuroinflammation in AD [4,5]. These activated cells switch from being neuroprotective with time and secrete proinflammatory cytokines, leading to neurotoxicity, neuronal dysfunction and death, thereby speeding up the progression of AD [58].

#### 2.2.4. Peripheral Inflammation and Infections

Inflammation linked to peripheral infections caused by pathogens (*Toxoplasma gondii*, *H. pylori*, *cytomegalovirus*), as well as chronic inflammatory diseases (obesity, diabetes, cardiovascular diseases), is a significant contributor to the pathophysiology of AD [72]. However, the possible mechanisms by which peripheral inflammation influences and promotes AD pathogenesis are quite complex and not well known, centering around pathogen-derived neurotoxic molecule composition, Aβ plaque formation, BBB disruption and impaired neurogenesis [72].

### 2.3. Therapeutic Implications

Interaction between Aβ plaque formation and the development of TAU pathology magnifies our understanding of the progression of AD and the refined contributions of the pathogenic TAU becomes imperative for therapeutic intervention development.

#### 2.3.1. Immunomodulation

Targeting peripheral inflammation has emerged as a potential therapeutic strategy for AD. Clinical trials investigating anti-inflammatory agents, such as nonsteroidal anti-inflammatory drugs (NSAIDs) and monoclonal antibodies, are underway [73].

#### 2.3.2. Anti-Microbial/Pathogenic Medications

Recent discoveries have shown that since Aβ presents anti-microbial properties, then there is a possibility of infectious etiology in AD hinting to infection-induced Aβ plaque formation characteristic in the disease pathology [74]. Weakened BBB and immune system are present in AD patients predisposing them to elevated risks of microbial infections that can cause chronic neuroinflammation, production of the possibly antimicrobial Aβ peptide, and neurodegeneration. Therefore, novel multiple diagnostic tests to detect major pathogens followed by anti-microbial treatments have emerged as a significant therapeutic intervention for AD [74,75,76].

#### 2.3.3. Precision Medicine

Advancements in understanding genetic factors associated with AD and peripheral inflammation may enable the development of personalized therapeutic approaches, tailoring treatment to an individual’s specific risk profile [77].

Peripheral inflammation is increasingly recognized as a critical player in the pathogenesis of AD. Understanding the relations between systemic inflammation, microglial activation and AD risk factors is providing clues to finding therapeutic interventions as well as prevention strategies. More research in this field will further enhance our understanding and help us to better deal with the grave neurodegenerative disease.

## 3. Pathogenesis and Risk Factors of Osteoarthritis and Possible Interplay with AD

### 3.1. Pathogenesis

#### 3.1.1. Definition/Introduction

Osteoarthritis (OA) is a prevalent and debilitating musculoskeletal disorder characterized by the progressive degeneration of articular cartilage, leading to joint pain and joint functional impairment. This part includes a comprehensive summary that examines the pathogenesis of OA and explores the multi-faceted risk factors involved and contributing to the development of this condition.

#### 3.1.2. Factors Contributing to the Pathogenesis of OA:

##### The Inflammatory Process in OA

A chronic low-grade inflammation plays an important and vital role in the pathogenesis of OA. Synovial inflammation, characterized by increased production of proinflammatory cytokines, such as IL-1β and TNF, contributes immensely to cartilage damage and pain [78]. Chronic inflammation is not exclusive to OA; it is also a hallmark feature of AD and is believed to contribute to neurodegenerative processes [4,5,79]. This shared element of chronic inflammation implies a plausible connection between OA and AD through the commonality of inflammatory pathways.

##### Articular Cartilage Degeneration

Another primary hallmark of OA is the breakdown of articular cartilage, which serves as a cushioning and lubricating structure within joint spaces in the body. Imbalances in cartilage homeostasis, including increased catabolism and diminished anabolism, lead to cartilage degeneration—a main feature of OA [80].

##### Mechanical Stress

Abnormal mechanical loading on the joints, resulting from factors such as joint misalignment or obesity, can also lead to mechanical stress on the articular cartilage. This stress is not only associated with cartilage and bone breakdown but also accompanied by the breakdown and activation of intracellular signaling pathways that are implicated in OA [81].

### 3.2. Modifiable and Non-Modifiable Risk Factors for Osteoarthritis Development

#### 3.2.1. Obesity

Excess body weight is a well-established risk factor for OA, particularly in the weight-bearing joints like knees and hips. Obesity can contribute to the pathology of OA through both mechanical and metabolic pathways. Importantly, chronic inflammation observed in obesity may be involved in the pathogenesis of not only OA, but also AD [82,83]. These links highlight the likelihood of the interaction between obesity, OA and AD, as well as illustrate the need for an integrated approach to engaging these complex health issues.

#### 3.2.2. Joint Overuse and Injuries

Athletes are mostly at risk of developing OA later in life from recurrent overuse or trauma to joints. Worn down or joint overuse can accelerate the mechanical destruction of cartilages, and if lesions or injuries are involved, then the damage becomes long term [80].

#### 3.2.3. Hormones and Arthritis

Among the many different factors in OA, the hormonal effect is what gets the most attention from researchers [84]. The onset of post-menopause, as the female reproductive life-cycle draws to an end and estrogen levels take a sudden plunge, amounts to a connection with increased susceptibility for developing OA. This also gives rise to intriguing questions on how cartilage is regulated by hormones and if it can break down like other tissues do. But as research into AD progresses, a more inclusive picture is developing. It has been discovered that sex hormones, particularly estrogen, influence AD pathogenesis [85]. Indeed, a colony of studies sheds light on these connections at every turn and mostly highlights the association between decreased estrogen and rising AD risk [86,87].

#### 3.2.4. Aging

Aging is an expressive risk factor for OA because of the reduced capability to repair the joint’s cartilage. In a survey with participants older than 60 years old, more than one-third revealed radiographic knee OA. Symptomatic knee OA was observed in about 4.9% and 16.7% of participants older than 26 and 45 years, respectively [88]. Another study demonstrated that the lifetime risk of developing symptomatic knee OA is 44.7%, with obesity enhancing this risk to approximately 60% in the case of a body mass index higher than 30 [89]. Importantly, aging is the most important risk factor for AD [80,90]. A recent study has demonstrated an incidence rate of four patients per 1000 people aged 65–69. This number increases drastically to 65 per 1000 for people aged 85–89 years [91].

#### 3.2.5. Race

Emerging research has shown that the prevalence, severity and patterns of OA can vary among racial and ethnic groups. Nevertheless, race plays a complicated role in the risk of developing OA. Some studies have shown that African-American individuals tend to have an enhanced prevalence of developing knee OA compared to their caucasian counterparts [92]. Race alone is not a direct cause of OA development; it is an important factor that modifies the frequency, severity and clinical manifestations of OA across various racial and ethnic groups.

The risk factors of OA surround a multiple spread of influences, with obesity standing out as a prominent contributor. Joint overuse and injuries can hasten joint degradation [93]. In postmenopausal women, estrogen level changes have become an important factor for developing OA risk. Aging also plays a key role in the pathogenesis of the disease, since it reduces the body’s effectiveness in joint cartilage repairs. Finally, genetic factors also can increase the vulnerability of developing this disease [94,95,96,97,98,99,100,101]. All these aforementioned factors are also involved in the pathogenesis of AD. Below, we describe some possible mechanisms that link OA and AD.

## 4. AD Development and OA: An Extensive Insight

AD and OA are two common age-related diseases that both bring about a multitude of distress and significant morbidity in affected patients. Research lately shows that a hidden link seems to exist between these two definite diseases. This review provides a detailed examination of the link between AD and OA development, drawing clear comprehension from within the studies through research and meta-analysis.

### 4.1. The OA–Dementia Link

#### 4.1.1. Animal Experimental Models

In a lead study conducted by Gupta et al. [102] to analyze the association between knee OA and AD on 5xFAD transgenic mice models, induction of OA in the AD mice led to a more exacerbated symptom in certain brain regions, increased inflammation including elevated IL-1β and TNF levels, and more Aβ disposition. This study helps us understand how OA development can exacerbate pathologies related to AD development.

Recently, it has been shown that concurrence of OA and AD appeared to build more neuroinflammation, accumulated Aβ deposition and heightened cognitive deficits in mice [103]. This study demonstrated that once OA is induced in the knees and temporomandibular joints of the Col1-IL1βXAT mice model, it triggers astrocyte and microglia activation, leading to the upregulation of inflammation-related gene expression. The study provides strong evidence that the development of AD occurs due to OA-induced peripheral and brain inflammation, suggesting an association between both pathologies.

#### 4.1.2. Biological Mechanisms

##### Inflammation as A Common Theme

Persistent peripheral inflammation that occurs in the joints during OA is accompanied by elevated levels of inflammatory mediators IL-1β, IL-6 and TNF [78,79]. These mediators are also attributed to the AD pathogenesis [79]. In OA, the inflammatory mediators produced by the immune cells are responsible for the degradation of the interstitial tissue and the production and release of many proteolytic enzymes, such as matrix metalloproteinases that decompose articular cartilage [104]. For example, during the pathophysiological process occurring in the course of OA, TNF secreted by the cells of the joint has the ability to cause local loss of articular cartilage as it binds to the TNF-R1 receptor [105], leading to chondrocyte death and a disorderly migration of chondrogenic progenitor cells (CPCs). This strips the cartilage of any possibility of regeneration, which exacerbates OA [106,107,108,109]. In addition, this cytokine also activates osteoclasts and chondrocytes [110], which contributes to the heightened production and release of proteolytic enzymes [111]. However, the effect of TNF concurs with the action of IL-1β in many cases of cartilage arthropathies including OA [112], hinting at evidence of synergistic interactions between these cytokines.

Further affirmation of the coaction of inflammatory mediators in OA pathogenesis brings into the limelight the actions of progranulin, a known growth factor with anti-inflammatory and immunomodulatory properties [113]. This molecule has a good ability to bind to TNF receptors [114], and is a natural antagonist that interferes with the TNF receptor signaling pathway [114]. Progranulin is significantly elevated in patients with OA during the course of the disease [115], possibly countering the effect of the TNF-R1 receptor signaling pathway. Therefore, following its actions, it is proffered that an imbalance between TNF and progranulin can rapidly progress or inhibit the development of OA [104].

These proinflammatory cytokines also have important roles in the pathogenesis of AD. It has been shown that TNF, IL-1β, IL-6 and IL-18 could either cross the BBB [52,53] or be produced by glial cells during AD [48,116] and eventually lead to tau hyperphosphorylation and neuronal loss [117]. TNF induces neuronal cell death [117,118,119] and impairs the phagocytosis of Aβ plaques [120].

The study by Walker et al. [121] examining the correlation between peripheral inflammatory injuries and the onset of AD discovered that systemic inflammation, which can come from infections or long-term illnesses, increases the likelihood of developing the disease. This causes inflammatory-linked responses in the brain, which involves microglia and astrocytes becoming active and releasing proinflammatory molecules. In addition, inflammation in the periphery can weaken the BBB, allowing inflammatory mediators to cross into the brain parenchyma. The breaking down of the barrier increases neuroinflammation and increases the rapid buildup of TAU pathology and Aβ. The study also highlights the clinical importance of these findings, indicating that controlling peripheral inflammation may have the potential to decrease AD risk and slow down the progression of the disease. To provide direction for future research, the study emphasizes the necessity of carrying out additional explorations into the intricate interplay between peripheral inflammation and AD, in order to discover potential therapeutic targets in this context [121].

Recently, a study including 374 AD patients demonstrated that OA was associated with a faster Aβ and TAU accumulation in primary motor and somatosensory brain regions [122]. In this sense, a preclinical study has also demonstrated that TAU is upregulated in an animal model of OA through the injection of collagenase. TAU deficiency also reduced the inflammatory response and damage induced by collagenase [123].

As discussed above, AD pathology can consequently lead to chronic inflammation. This process aggravates the Aβ deposition and hyperphosphorylated TAU aggregation [124,125]. Yet fascinatingly, research also reveals synergistic actions between Aβ and proinflammatory cytokines to promote pathology associated with AD [126]. Therefore, the inflammatory mediators produced during OA may lead to the maintenance of the neuroinflammation and neurodegenerative process by activating glial cells and enhancing deleterious factors.

##### The Role of APOE Variants

In a thorough study by van den Bosch et al. [101], researchers looked at how the APOE gene and its isoforms, especially APOE-ε4 and APOE-ε3, can influence OA. Their study focused on the severity of OA joint pathology, showing a possible connection between APOE gene variants and the development of OA. The study found a big difference between mice with APOE-ε4 and APOE-ε3, with the APOE-ε4 mouse having worse joint damage due to OA. This interesting finding suggests that the APOE-ε4 genetic disposition might be a huge factor in developing inflammatory OA [101]. Importantly, APOE-ε4 gene expression is related to a higher risk of AD. More studies of this gene link will help enlighten us as to how genes and the risk of OA are connected. This can give new knowledge about understanding and controlling this ailment—OA, which is possibly also linked to AD.

##### Common miRNA Alterations in AD and OA

As discussed previously, miRNA may be involved in the pathogenesis of AD. Interestingly, as it happens in AD brain tissues, there is dysregulation of the miRNAs 9, 16, 101, 124, 132 and 146 in OA [12]. For example, miRNA9 and miRNA124 may have protective and deleterious roles in OA, respectively [127,128]. Besides, it has been shown that inhibition of miRNA146 reduced the destruction of the articular cartilage in an animal model of OA [129]. On the other hand, inhibition of miRNA132 increased inflammatory factors in chondrocytes [130]. Thus, it is possible that these miRNAs may contribute to the progression of both OA and AD, although these alterations in both conditions could represent only an epiphenomenon. Therefore, further studies are necessary to investigate the possible role of specific miRNAs as a link between OA and AD.

#### 4.1.3. Epidemiological Insights, Observational Studies, Systematic Reviews and Meta-Analysis

A nationwide cohort study carried out in Taiwan showed that people with OA were found to have a higher chance of developing dementia [131]. Another comprehensive study by Ikram et al. [132] utilized data from the Medical Expenditure Panel Survey (MEPS) during years ranging from 2009 to 2015 to investigate how OA incidence in older people aged 65 or more might be connected to both long-lasting pain (PIA) and AD-related dementias (ADRDs). The study showed that adults aged over 65 years who reported experiencing PIA with or without OA, had significantly higher chances of developing ADRD in comparison to those subjects without OA or PIA. The results of this study using a substantial sample of older adults in the MEPS emphasized the notable impact of OA and persevering pain on the increased risk of AD and related dementias [132]. In yet another study, Weber et al. [133] showed a strong connection between OA and a higher chance of occurrence of dementias. The systemic review and meta-analysis included six studies, which were observational and composed of 388,252 individuals where they checked if OA was linked to increased risk for developing dementia. The results showed a clear link between OA and the increased risk of dementia.

A prospective study investigated how chronic knee pain and lower back pain are linked with the chances of developing dementia. It was shown that long-lasting joint pain might be linked to an increased risk of dementia. This made the investigators more interested in studying the ties between OA and AD [134]. The inflammatory process may spread from joints to the brain and contribute to neuroinflammation [103]. Moreover, pain caused by OA could indirectly affect an individual’s cognitive health because it predisposes to social isolation/loneliness and lessens participation in mentally stimulating activities or tasks [135].

In yet another study by Innes and Sambamoorthi [136], the authors intricately investigated the association and connection between OA, the burden of related pain and the incidence of AD-related dementia (ADRD) in a retrospective cohort study whilst focusing on US medicare beneficiaries (11 pooled cohorts, 2001–2013). The results of their study revealed that individuals/beneficiaries with OA and increased pain burden due to OA face an elevated risk of developing ADRD [136].

To sum up this part, a growing body of evidence from animal studies, epidemiological studies and meta-analyses, suggests a significant association between osteoarthritis and the risk of developing AD and its related dementias. Biological mechanisms such as chronic inflammation can provide ties between these conditions and further investigations are required to fully understand the complexities of these links; these discoveries will open new ways to explore possible prevention and treatment plans for both OA and AD by targeting the common links between them.

## 5. Perspectives on Potential Therapeutic and Preventive Strategies by Targeting Common Pathways between OA and AD

As explained by Weber et al. [133], the growing connection between OA and AD shows that both can have similar pathologic paths, knowing these shared pathogenic pathways between them brings clearly into the picture an opportunity to produce treatment plans that can transform, shape and cure these health conditions simultaneously [132,133]. This advanced perspective explores the potential interventions, drawing insights from recent research. Nevertheless, the prospective neuroprotective results of medicines used to treat symptoms of OA are a developing area of research interest. A handful of medicines and treatments regularly used for managing OA-related symptoms have been examined for their neuroprotective effects, especially in the reference frame of AD and its related neurodegenerative conditions. Below are some key cognizance into topics discussed previously.

### 5.1. Targeting Inflammation

#### 5.1.1. NSAIDs

Nonsteroidal inflammatory drugs, such as naproxen and ibuprofen, are recurrently used to manage inflammation and pain related to OA. Some research has advocated that prolonged NSAID use is correlated with a reduced risk of AD [94]. Nonetheless, prolonged NSAID use has side effects, which makes their long-term safety and effectiveness as neuroprotective agents remain a theme of ongoing scrutiny.

#### 5.1.2. Disease-Modifying Antirheumatic Drugs (DMARDs)—Immunosuppressive and Modulatory Agents

These are a class of drugs such as methotrexate and adalimumab, indicated for the remedy of inflammatory arthritides, including rheumatoid arthritis, psoriatic arthritis and ankylosing spondylitis [137,138,139]. Each of these drugs has a unique way of action essentially interfering with key fundamental pathways in the inflammatory cascade [140]. For this reason, DMARDs could be investigated for their potential to regulate neuroinflammation in AD [121].

### 5.2. Disease-Modifying Osteoarthritis Drugs (DMOADs)

DMOADs are medications (anticytokine therapy, enzyme inhibitors, growth factors, gene therapy, peptides) that are currently in clinical and preclinical trials intended to be utilized to induce the repair and regeneration of articular tissues in OA pathology [141]. Although some trials with chondroitin sulfate and glucosamine to manage OA conditions have shown potential in preserving cognitive functions [142], their exact mechanisms in this aspect are still being researched because these substances are believed to possess antiinflammatory and neuroprotective properties and target common pathways between OA and neurodegenerative diseases like AD.

### 5.3. Statins

Statins are drugs frequently used as lipid-lowering medications to reduce cholesterol levels and have been researched for their potential effectiveness in neuroprotection. Current research has attributed statin usage to be associated with a potential reduction in the risk of dementia and developing AD [143]. Increased cholesterol levels have been associated with cognitive decline and AD development via various pathways and mechanisms like Aβ accumulation, dysfunction of the BBB, oxidative stress, vascular compromise, neuroinflammation, tau hyperphosphorylation and brain parenchyma structural changes [144]. Therefore, superintending cholesterol levels is key in dealing with the possible likelihood of AD and cognitive impairment [143]. With more investigations needed to totally comprehend the outcomes of statins on AD, the findings mentioned above provide a worthy standpoint for examining protective measures and therapeutic interventions against the backdrop of risk factors associated with vascular conditions and AD development.

### 5.4. Glucocorticoids

Studies and research by Calsolaro and Edison [145] have revealed the role of chronic inflammation in AD development, which is also fostered partially by unbridled immune system response in the brain, thus making the subject area pertaining to the beneficial anti-inflammatory properties of glucocorticoids captivating, although the relationship between inflammation and AD development is quite composite. This uncharted territory provokes an intended assessment of the utilization of glucocorticoids in this course, but consideration must be taken as their predisposition for prolonged or high-dose administration may unknowingly aggravate the path of cognitive decline and the underlying pathological hallmarks of AD. However, research into developing selective glucocorticoid receptor modulators (SGRMs) with the intention of channeling the anti-inflammatory properties of these drugs and at the same time minimizing the side effects, especially for the therapy of neurodegenerative conditions, has been explored. SGRMs are directed to target specific glucocorticoid receptor variants in order to achieve more precise control of the immune system and attenuate the deleterious effect linked to prolonged glucocorticoid use [146].

### 5.5. Molecular Approaches

#### Shared Pathways

The shared links between OA and AD provide optimistic approaches to the generation of selected treatments. One of such shared links is a pathway known as the Wnt signaling pathway, a molecular signaling pathway consisting of Wnt proteins, a family of secreted lipoproteins that activate different intracellular signaling pathways by binding to several receptors and co-receptors at the cell surface [145]. The functions of this signaling pathway are crucial to cellular processes such as leading to changes in gene expression, cytoskeleton reorganization and tissue development, maintenance and repairs [147,148]. Meanwhile, Wnt signaling specifically has been attributed to cartilage and bone homeostasis, with its abnormal regulation advancing abnormal tissue remodeling in OA, as well this pathway influences AD pathogenesis since it’s required for neuronal synaptic plasticity/maintenance, regulation/clearance of Aβ production and and synaptic alteration during aging [149,150,151,152].

Abnormal Wnt signaling has been noticed in AD-affected brain samples, uplifting the likelihood of treatment interventions earmarking these pathways [153]. By revealing the combined involvement of Wnt signaling in both OA and AD pathogenesis, investigators can study and engage in the development of molecules modulating this pathway that will bring about innovative therapeutic breakthroughs in this aspect and will simultaneously address both OA and AD pathophysiology and offer a multifaceted elucidation to these complex health conditions.

### 5.6. Personalized Interventions—Precision Medicine

Precision medicine (PM), also known as “personalized medicine” is an unconventional approach to mending disease prevention and treatment, taking into account differences in an individual’s genetic makeup, environment and lifestyle, with the goal of targeting the right treatments to the right patients at the right time [154]. In this context, PM offers a potential strategy that can be made to handle and tackle the shared paths between OA and AD with the key approach of identifying and confronting the innumerable contributors for both conditions in each patient.

Using information about genes and the body’s metabolic levels, health care researchers can make special interventions, foresee risks [155], pick treatments that work best for certain people [156] and continuously observe/monitor a patient’s condition and recovery [157]. This approach holds the potential to revolutionize the management of OA and AD by addressing their underlying causes in a highly individualized manner. Also, PM uses modern research technology to pinpoint important genetic indicators and metabolic routes linked with neuroinflammation [158]. As understandings widen more about these health conditions, precise aimed actions and treatments can be made, giving hope for better cures or preventions.

These findings suggest that the drugs and therapies for managing OA symptoms might be neuroprotective to the brain, but it is important to underscore that more studies are required to prove these relationships clearly alongside the needed information about how safe these medicines are and their effects over time when used in managing neurodegenerative diseases.

Individuals concerned about their cognitive health should work on a complete lifestyle change, like eating balanced meals [45], being active with exercise [159], doing mental challenges and routine medical examinations, as these are important in keeping the mind healthy and help prevent diseases that affect cognition and memory such as AD and other neurodegenerative conditions

The growing knowledge of shared routes between OA and AD is giving hope for novel ways to prevent or treat the condition. Targeting inflammation, changing our way of life (healthy lifestyle) and traversing molecular and genetic approaches can open new routes for innovative therapies.

## 6. Preliminary Conclusions

### 6.1. Unraveling the Interplay between OA and AD

The complex connection between OA and AD is a very interesting health issue that needs to be studied and researched more. This review has shown many mixed links and given important thoughts about the interaction between these seemingly different distinct health conditions.

#### 6.1.1. Shared Pathogenic Mechanisms

At its core, the OA–AD association finds common ground in shared pathogenic mechanisms with chronic inflammation being a pivotal player, as well as an emerging connecting thread in both disease conditions. Peripheral inflammation, often associated with OA, extends its influence into the area of AD. Nevertheless, the potentiality of systemic inflammation to break down the BBB and stimulate neuroinflammatory responses in the brain parenchyma opens a fascinating avenue for further research.

#### 6.1.2. Epidemiological Insights

Meta-analysis and epidemiological studies emphasize the interconnection between pain in OA and AD related dementias; this evidence hints at the role and importance of pain interference in the development of AD, making pain management in OA a potential target for mitigating AD risk.

#### 6.1.3. Biological Insights

Animal models have further solidified the link between OA and AD. Findings demonstrate that OA can accelerate Aβ deposition and neuroinflammation, thus unraveling the biological intricacies of their connection.

#### 6.1.4. Prospective Therapeutic and Preventive Strategies

Fascinatingly, medicines and therapies regularly used to manage the symptoms of OA have emerged as prospective contenders for neuroprotection in AD therapy, such drugs like NSAIDs, disease-modifying OA drugs, statins and glucocorticoids, have been carefully examined for their potential neuroprotective effects, bringing a possibility of alleviating OA symptoms and at the same time preventing and possibly safeguarding against neuroinflammation in AD.

#### 6.1.5. Precision Medicine and Genetic Approaches

Interventions made based on a person’s genes and metabolic requirements hold promising chances as prospective treatments. The APOE-ε4 gene variant, found involved in both conditions of OA and AD, is an important center point for thorough investigation and research. It is possible the gene variant helps cause inflammatory arthritis, underscoring the need and prospect to study the genes that may help us understand better why OA develops and how it happens. Nevertheless, accurate state-of-the-art gene and molecular intervention actions in this direction might lead to new treatment approaches.

## 7. Conclusions

As described before, various factors contribute to the production of inflammatory mediators during OA. These mediators lead to a plethora of pathological events in the brain, such as further inflammation, aggregation of Aβ, hyperphosphorylation of tau, synaptic dysfunction and neurodegeneration (Figure 1). Therefore, potential interventions that reduce the pathogenesis of OA, especially in the early phases of the disease, could also avoid the development of AD. Nonetheless, it is important to note that most studies cited in this review are observational and preclinical studies, which represents a limitation in this field.

In simple words to condense the discussion, the link between OA and AD is a topic ready for further research. The search to understand their common paths and find ways for prevention and treatment is an exciting area. Collaborative research works across different areas and disciplines will be important in illuminating this complicated connection and bringing hope for a better life quality for individuals experiencing both health issues. The expedition goes on, led by the search for more understanding and hope for a healthier future and better health ahead for all.

## Figures and Tables

**Figure 1 ijms-25-03044-f001:**
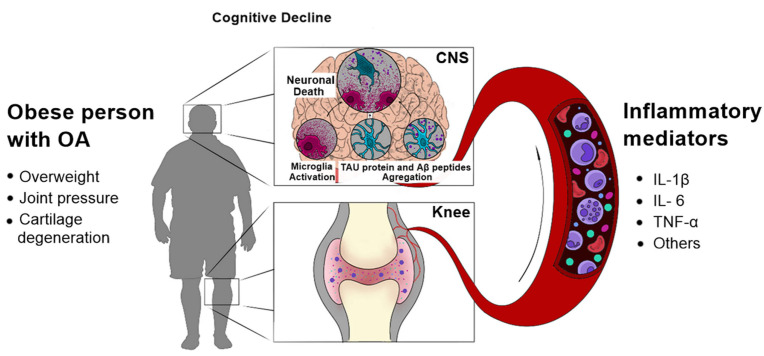
Possible molecular mechanisms linking OA with AD. A variety of mediators, released from inflamed tissues during OA, leads to pathological events in the CNS, such as aggregation of Aβ peptides and hyperphosphorylated tau, and neuroinflammation. These events could, in turn, initiate or contribute to the progression of AD.

## Data Availability

Not applicable.

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
