# Peer review of "Molecular Mechanisms Linking Osteoarthritis and Alzheimer’s Disease: Shared Pathways, Mechanisms and Breakthrough Prospects"

_ijms, 2024, doi:10.3390/ijms25053044_

Round 1
Reviewer 1 Report
Comments and Suggestions for Authors
Manuscript ID: ijms-2827302
Type of manuscript: Review
Title: Molecular mechanisms linking Osteoarthritis and Alzheimer’s Disease; Shared Pathways, Mechanisms, and Breakthrough Prospects
The theme of the literature review undertaken by the authors seems interesting; it searches molecular mechanisms, factors that link Alzheimer’s disease with osteoarthritis addressing also future perspectives and innovative therapies of both diseases.
I have some important comments that can improve the work and make it more acceptable to the potential reader.
1. Abstract; line 11, add “as” after such
2. Line 12-16; This sentence is incomprehensible. What is the relationship between genetic and environmental factors and the fact that cognitive changes in AD are mild at first and progress over time? I don't know what the authors had in mind. Please rewrite this.
3. Line 136, please explain shortcut LOAD when it appears in the text for the first time.
4. Section 2.1. Peripheral inflammation and AD Pathogenesis. The chapter is very laconic. I would expect more detailed information on how systemic inflammation affects the destruction of the blood brain barrier.
Also, as you wrote : ”Also, there is evidence suggesting that the disintegrity of BBB may be involved in neurodegenerative illnesses including Alzheimer's disease (Nation et al., 2019; Zhao et al., 2015).” Please give more details on what the evidence is.
5. The situation is similar with the other chapters. The authors signal some connections, but do not go into details and mechanisms, even hypothesized. it's important to complete this.
6. I also have the impression that the work should be corrected by someone fluent in English.
Comments on the Quality of English LanguageCorrection of the text by someone fluent in English would improve its readability.
Author Response
Manuscript ID: ijms-2827302
Type of manuscript: Review
Title: Molecular mechanisms linking Osteoarthritis and Alzheimer’s Disease; Shared Pathways, Mechanisms, and Breakthrough Prospects
The theme of the literature review undertaken by the authors seems interesting; it searches molecular mechanisms, factors that link Alzheimer’s disease with osteoarthritis addressing also future perspectives and innovative therapies of both diseases.
I have some important comments that can improve the work and make it more acceptable to the potential reader.
Response: We thank the reviewer for his/her important comments.
- Abstract; line 11, add “as” after such
Response: In the new version of the manuscript, we have improved the text from line 10 to 19. Therefore, this part was rephrased.
- Line 12-16; This sentence is incomprehensible. What is the relationship between genetic and environmental factors and the fact that cognitive changes in AD are mild at first and progress over time? I don't know what the authors had in mind. Please rewrite this.
Response: As stated in question #1, we have improved the text from line 10 to 19. Therefore, this part was rephrased.
- Line 136, please explain shortcut LOAD when it appears in the text for the first time.
Response: We have written it in full (page 3).
- Section 2.1. Peripheral inflammation and AD Pathogenesis. The chapter is very laconic. I would expect more detailed information on how systemic inflammation affects the destruction of the blood brain barrier.
Response: We have expanded this part in the new version of the manuscript.
Also, as you wrote : ”Also, there is evidence suggesting that the disintegrity of BBB may be involved in neurodegenerative illnesses including Alzheimer's disease (Nation et al., 2019; Zhao et al., 2015).” Please give more details on what the evidence is.
Response: We have given more details about the evidence provided by the articles.
- The situation is similar with the other chapters. The authors signal some connections, but do not go into details and mechanisms, even hypothesized. it's important to complete this.
Response: In the new version of the manuscript, we have included the links between the possible mechanisms of osteoarthritis and Alzheimer's disease.
- I also have the impression that the work should be corrected by someone fluent in English.
Response: We have extensively revised the English.
Comments on the Quality of English Language
Correction of the text by someone fluent in English would improve its readability.
Response: We have extensively revised the English.
Reviewer 2 Report
Comments and Suggestions for Authors
Reviewer comments
This review article discusses molecular mechanisms that link to major aging diseases Osteoarthritis and Alzheimer’s disease. The MS has discussed the basic mechanism of the pathology of OA and AD that are possibly linked or share the same mechanisms of disease, including inflammation. To keep this point in mind, authors have discussed and concluded the common therapeutic targets and agents that can treat both OA and AD.
The paper is not written in an organized manner and has many shortcomings. The MS has very basic information (too much information in a very simple way that is not needed) and does not provide potent research-based data to show the molecular mechanisms that link OA and AD.
Scientific comments
1. In the subsection, Dysregulated miRNAs in AD; you have not provided the name of any specific miRNAs. It is a very general statement. Provide some scientific data that could be linked to
2. In the subsection, Global prevalence; references are old. Give the current reference for AD epidemiology.
3. In the subsection, Global variation; the reference is of 2003. It's 20 years old. Provide a reference to the current scenario.
4. Revise the epidemiology section.
5. Provide a reference for lines 165-169.
6. Provide specific references for lines 331-342.
7. Provide specific references for lines 474-479.
8. Line 357, ‘Several studies have unveiled a strong link between OA and AD’; you have written several studies however from lines 357-374 you have discussed only one research article (Kyrkanides et al., 2011) of 2011.
9. Line 591-596, the cited reference has not discussed the association between OA and AD, and not about precision medicine. Check the reference and correct it.
10. There are two citations for Heneka et al., 2015; it is confusing which one is discussed and cited.
11. Discuss the following papers-
· Rheumatoid arthritis and risk for Alzheimer’s disease: a systematic review and meta-analysis and a Mendelian Randomization study.
· The Association of Osteoarthritis and Related Pain Burden to Incident Alzheimer’s Disease and Related Dementias: A Retrospective Cohort Study of U.S. Medicare Beneficiaries.
Minor comments
1. Check the citation pattern of the journal and change it accordingly.
2. Check the reference style on the journal website and correct it accordingly.
3. Line 71-72; check the pattern of writing.
Typo error
1. The MS has many typo errors in the font style, full stop, colon, semicolon etc.
Comments on the Quality of English LanguageMinor editing of the English language is needed. Typo errors are too many throughout the MS.
Author Response
Reviewer comments
This review article discusses molecular mechanisms that link to major aging diseases Osteoarthritis and Alzheimer’s disease. The MS has discussed the basic mechanism of the pathology of OA and AD that are possibly linked or share the same mechanisms of disease, including inflammation. To keep this point in mind, authors have discussed and concluded the common therapeutic targets and agents that can treat both OA and AD.
The paper is not written in an organized manner and has many shortcomings. The MS has very basic information (too much information in a very simple way that is not needed) and does not provide potent research-based data to show the molecular mechanisms that link OA and AD.
Response: We thank the reviewer for the revision of the manuscript. In this new version, we have improved the text and provided more information on the mechanisms that link OA and AD.
Scientific comments
- In the subsection, Dysregulated miRNAs in AD; you have not provided the name of any specific miRNAs. It is a very general statement. Provide some scientific data that could be linked to
Response: In the new version of the manuscript, we have added specified the miRNAs in AD and provided some information on their roles in both OA and AD.
- In the subsection, Global prevalence; references are old. Give the current reference for AD epidemiology.
Response: We have included current references.
- In the subsection, Global variation; the reference is of 2003. It's 20 years old. Provide a reference to the current scenario.
Response: We have included current references.
- Revise the epidemiology section.
Response: Epidemiology section revised and references added.
- Provide a reference for lines 165-169.
Response: References were provided in the new version.
- Provide specific references for lines 331-342.
Response: References were provided in the new version.
- Provide specific references for lines 474-479.
Response: References were provided in the new version.
- Line 357, ‘Several studies have unveiled a strong link between OA and AD’; you have written several studies however from lines 357-374 you have discussed only one research article (Kyrkanides et al., 2011) of 2011.
Response: The statement ‘Several studies‘ was corrected.
- Line 591-596, the cited reference has not discussed the association between OA and AD, and not about precision medicine. Check the reference and correct it.
Response: The reference was checked and the statement was revised.
- There are two citations for Heneka et al., 2015; it is confusing which one is discussed and cited.
Response: The two citations have been properly distinguished in the text.
- Discuss the following papers-
- Rheumatoid arthritis and risk for Alzheimer’s disease: a systematic review and meta-analysis and a Mendelian Randomization study.
- The Association of Osteoarthritis and Related Pain Burden to Incident Alzheimer’s Disease and Related Dementias: A Retrospective Cohort Study of U.S. Medicare Beneficiaries.
Response: We have discussed both papers in the new version of the review.
Minor comments
- Check the citation pattern of the journal and change it accordingly.
Response: We prepared the manuscript according to the pattern of the journal.
- Check the reference style on the journal website and correct it accordingly.
Response: We prepared the manuscript according to the pattern of the journal.
- Line 71-72; check the pattern of writing.
Response: We have corrected it.
Typo error
- The MS has many typo errors in the font style, full stop, colon, semicolon etc.
Response: We have extensively revised the text.
Comments on the Quality of English Language
Minor editing of the English language is needed. Typo errors are too many throughout the MS.
Response: We have extensively revised the text.
Reviewer 3 Report
Comments and Suggestions for Authors
The review paper provides a comprehensive overview of the molecular mechanisms linking Osteoarthritis and Alzheimer’s Disease; Shared Pathways, Mechanisms, and Breakthrough Prospects. The review aims to Investigate the possible links between two prevalent age-related diseases – OA and AD – that hold significant research potential. The introduction effectively outlines the key characteristics of both OA and AD, highlighting their shared risk factors and potential co-occurrence. The planned exploration of molecular mechanisms, connections, and prospects creates a clear roadmap for the review.
The authors have done a good job of synthesizing information from various sources and presenting it in an organized and readable format. However, there are a few points that could be improved or clarified:
- Strengthen the argument for co-occurrence: While acknowledging independent factors for both diseases, delve deeper into specific evidence suggesting OA potentially acts as a risk factor for AD. Cite relevant studies with statistical data and explore possible underlying mechanisms.
- Clarify the scope of the review: Specify the types of molecular mechanisms you plan to address. Is it solely protein aggregation, inflammatory pathways, or other shared biological processes? Narrowing the scope might provide a more focused and impactful analysis.
- Consider alternative perspectives: Acknowledge existing controversies or alternative explanations for the observed correlation between OA and AD. Discuss limitations of previous research and address possible confounding factors.
- Balance optimism with realistic limitations: While presenting prospects and innovative treatments, avoid overpromising breakthroughs. Emphasize the need for further research and acknowledge the challenges in translating basic science discoveries into clinical applications.
- Improve flow and conciseness: Streamline the introduction by avoiding redundancy and condensing unnecessary details. Maintain a clear logical progression throughout the review.
- Enrich references: Update and expand the reference list to include recent key publications and authoritative reviews in the field.
- You could consider including a section on current clinical trials or drug development efforts targeting shared pathways between OA and AD.
- Explore the potential impact of early OA diagnosis and intervention on preventing or delaying the onset of AD.
- Discuss ethical considerations of investigating shared mechanisms between two complex and heterogeneous diseases.
By addressing these suggestions and further refining your review, you can produce a valuable contribution to the understanding of the intriguing link between OA and AD, paving the way for future research and potential therapeutic advancements.
Author Response
The review paper provides a comprehensive overview of the molecular mechanisms linking Osteoarthritis and Alzheimer’s Disease; Shared Pathways, Mechanisms, and Breakthrough Prospects. The review aims to Investigate the possible links between two prevalent age-related diseases – OA and AD – that hold significant research potential. The introduction effectively outlines the key characteristics of both OA and AD, highlighting their shared risk factors and potential co-occurrence. The planned exploration of molecular mechanisms, connections, and prospects creates a clear roadmap for the review.
The authors have done a good job of synthesizing information from various sources and presenting it in an organized and readable format.
Response: Thank you very much for revising the manuscript and providing paramount suggestions. We believe we have significantly improved the review.
However, there are a few points that could be improved or clarified:
1. Strengthen the argument for co-occurrence: While acknowledging independent factors for both diseases, delve deeper into specific evidence suggesting OA potentially acts as a risk factor for AD. Cite relevant studies with statistical data and explore possible underlying mechanisms.
Response: In the new version of the manuscript, we have provided information of the possible links between OA and AD.
2. Clarify the scope of the review: Specify the types of molecular mechanisms you plan to address. Is it solely protein aggregation, inflammatory pathways, or other shared biological processes? Narrowing the scope might provide a more focused and impactful analysis.
Response: We intended to discuss different mechanisms that may link OA and AD. These pathways may also be connected in both diseases. We believe we have narrowed the scope in the current version of the review.
3. Consider alternative perspectives: Acknowledge existing controversies or alternative explanations for the observed correlation between OA and AD. Discuss limitations of previous research and address possible confounding factors.
Response: We have discussed the limitations and confounding factors.
4. Balance optimism with realistic limitations: While presenting prospects and innovative treatments, avoid overpromising breakthroughs. Emphasize the need for further research and acknowledge the challenges in translating basic science discoveries into clinical applications.
Response: We emphasized the limitations of the current literature and the perspectives in the field.
Improve flow and conciseness: Streamline the introduction by avoiding redundancy and condensing unnecessary details. Maintain a clear logical progression throughout the review.
Response: We have made the current version more concise.
5. Enrich references: Update and expand the reference list to include recent key publications and authoritative reviews in the field. You could consider including a section on current clinical trials or drug development efforts targeting shared pathways between OA and AD.
Response: In the new version of the manuscript, we have added new references that provide information and strength the link between AD and OA.
6. Explore the potential impact of early OA diagnosis and intervention on preventing or delaying the onset of AD.
Response: Along the text, we have included information on the possible links between OA and AD, as well as pharmacological targets in the prevention or therapeutic of AD.
7. Discuss ethical considerations of investigating shared mechanisms between two complex and heterogeneous diseases.
Response: We were not sure about this point required by the reviewer. However, we emphasize that we have discussed the complexity of both diseases and possible paths to investigate potential therapies.
By addressing these suggestions and further refining your review, you can produce a valuable contribution to the understanding of the intriguing link between OA and AD, paving the way for future research and potential therapeutic advancements.
Response: We thank the reviewer for the important suggestions. We believe we have improved the review.
Round 2
Reviewer 1 Report
Comments and Suggestions for Authors
The article was corrected according to my comments, so I have no further comments, I am satisfied.
Reviewer 2 Report
Comments and Suggestions for Authors
All the suggested comments has been incorporated by author in revised paper. Paper can be accepted in present form.
Reviewer 3 Report
Comments and Suggestions for Authors
We appreciate the authors' extensive revisions, which significantly improved the manuscript's quality. They have successfully addressed all concerns raised by the reviewers, and we agree that the manuscript is now acceptable for publication in its current form.